# Wildfire smoke knows no borders: Differential vulnerability to smoke effects on cardio-respiratory health in the San Diego-Tijuana region

**Lara Schwarz**[1,2]\*, **Rosana Aguilera**[3], **L. C. Aguilar-Dodier**[4], **Javier Emmanuel Castillo Quiñones**[4], **María Evarista Arellano García**[5], **Tarik Benmarhnia**[3]

**1** School of Public Health, San Diego State University, San Diego, CA, United States of America, **2** Herbert Wertheim School of Public Health and Longevity Science, University of California, San Diego, La Jolla, CA, United States of America, **3** Scripps Institution of Oceanography, University of California, San Diego, La Jolla, CA, United States of America, **4** Facultad de Ciencias Químicas e Ingeniería, Universidad Autónoma de Baja California, Tijuana, México, **5** Facultad de Ciencias, Universidad Autónoma de Baja California, Mexicali, México

\* lnschwar@health.ucsd.edu

**Data Availability Statement:** Code and datasets that are shareable are provided in the following repository: https://github.com/benmarhnia-lab/

## Abstract

Exposure to fine particles in wildfire smoke is deleterious for human health and can increase cases of cardio-respiratory illnesses and related hospitalizations. Neighborhood-level risk factors can increase susceptibility to environmental hazards, such as air pollution from smoke, and the same exposure can lead to different health effects across populations. While the San Diego-Tijuana border can be exposed to the same wildfire smoke event, socio-demographic differences may drive differential effects on population health. We used the October 2007 wildfires, one the most devastating wildfire events in Southern California that brought smoke to the entire region, as a natural experiment to understand the differential effect of wildfire smoke on both sides of the border. We applied synthetic control methods to evaluate the effects of wildfire smoke on cardio-respiratory hospitalizations in the Municipality of Tijuana and San Diego County separately. During the study period (October 11th- October 26th, 2007), 2009 hospital admissions for cardio-respiratory diseases occurred in San Diego County while 37 hospital admissions were reported in the Municipality of Tijuana. The number of cases in Tijuana was much lower than San Diego, and a precise effect of wildfire smoke was detected in San Diego but not in Tijuana. However, social drivers can increase susceptibility to environmental hazards; the poverty rate in Tijuana is more than three times that of San Diego. Socio-demographics are important in modulating the effects of wildfire smoke and can be potentially useful in developing a concerted regional effort to protect populations on both sides of the border from the adverse health effects of wildfire smoke.

border_smoke_cardio_resp.git Data on hospitalizations (patient discharge data) in California is confidential and cannot be made publicly available but can be requested from the California Department of Health Care Access and Information at the following link: https://hcai.ca.gov/data-and-reports/research-data-request-information/.

**Funding:** This work was supported by the Fogarty International Center of the National Institutes of Health (NIH) and the University of California Global Health Institute (UCGHI) (D43TW009343 to LS). The content is solely the responsibility of the authors and does not necessarily represent the official views of the NIH or UCGHI. The funders had no role in study design, data collection and analysis, decision to publish, or preparation of the manuscript.

**Competing interests:** The authors have declared that no competing interests exist.

## Introduction

The length, intensity, and severity of wildfires have increased under climate change [1] with record-breaking events occurring frequently in recent years. This contributes to high air pollution levels; the worst air quality in decades has been observed in Western North America these recent years due to raging wildfires [2]. Smoke plumes can extend far beyond the wildfire's perimeter and cause unhealthy air quality for neighboring cities, states, and even countries [3]. While borders are a physical barrier to human mobility, they provide no obstruction to the transport and flow of fine airborne particles; wildfire smoke can move freely, harming the health of residents on both sides.

The San Diego-Tijuana border (western coast of United States and Mexico) is a unique context to study the health effects of wildfire smoke due to its meteorological and socio-demographic characteristics. The region has been highly affected by climate change; in the Western United States (US), the amount of forest area burned from wildfires was more than 10 times greater from 2003–2012 than it was in 1973–1982 [4]. Furthermore, precipitation is expected to decrease by 50% in Southwest California and up to 75% in Baja California [5], increasing the risk of future wildfires in an already water-deprived region. San Diego and Tijuana cities are each located less than 20 miles from the border and less than 25 miles from each other so wildfires starting in either country may produce smoke that reaches both sides of the border. San Diego and Tijuana are two major cities, and make up the largest bi-national conurbation shared between the US and Mexico and the fourth-largest in the world [6]. San Diego County was estimated to have a population of 3 million while the Municipality of Tijuana was home to 1.6 million residents according to the 2010 census [7]. The region's susceptibility to climate impacts and high population density separated by an international border makes it a unique context to explore the differential public health manifestation of exposure to wildfire smoke.

Fine particulate matter ($PM_{2.5}$), one of the main components of wildfire smoke [8], is composed of inhalable particles that produce harmful health effects [9, 10]. $PM_{2.5}$ are particles under 2.5 microns in aerodynamic diameter that are small enough to go deep into the lungs and even enter the bloodstream and are consistently associated with increases in cardio-respiratory hospitalizations [11–13]. Additionally, in California, wildfires account for 50% of total primary $PM_{2.5}$ emissions, and this percentage is increasing under climate change [14]. Furthermore, wildfire-specific $PM_{2.5}$ is more harmful to respiratory hospital admissions than non-wildfire $PM_{2.5}$ in Southern California [15]. Previous research on the effect of the 2007 San Diego wildfires on Medi-Cal emergency department hospitalizations showed a 34% increase in respiratory visits and 112% in asthma-specific diagnoses during the peak fire period [16]; while a pronounced spatial heterogeneity in health effects was also observed across San Diego County [17]. Not all populations are equally susceptible to the effects of wildfire smoke, yet there is a scarcity of evidence of these impacts for populations living in low and middle-income contexts [18].

Although Tijuana and San Diego are close in proximity, they have contrasting socio-demographic profiles which may be important in driving differential susceptibility to wildfire smoke effects on health. Socio-economic status is well documented as an effect modifier in the association between air pollution and various health effects [19–21]. Socio-economic factors may increase susceptibility to air pollution impacts by increasing the risk of other conditions such as asthma and/or affecting development and resistance to other disease threats [22]. A study conducted in North Carolina (US) estimated that counties with a higher poverty level had twice the risk of having an emergency department visit following exposure to wildfire smoke than counties with lower poverty levels [23]. The same study showed that income inequality is a risk factor for increased emergency department visits due to smoke exposure [23].

Neighborhood-level risk factors faced by those living in low-income neighborhoods may increase their susceptibility to environmental hazards [24]. Stressors, such as crime, noise, and traffic can lead to acute and chronic changes in the functioning of body systems and increase the effect of exposures such as wildfire smoke [20]. Therefore, population-specific estimates of the epidemiological effects of wildfire smoke are critical in understanding its role in exacerbating adverse health outcomes and can contribute to informing preparedness and response to this increasingly prevalent environmental health hazard. We capitalize on a unique natural experiment in which smoke affected the entire San Diego-Tijuana region to assess if the same wildfire event impacted the two socio-economically diverse communities differentially. Although there are many studies estimating the health effects of wildfire smoke in San Diego and Southern California, to our knowledge no research has investigated these effects in Tijuana or Baja California (Mexico).

The aim of this study was to understand the differential effect of wildfire smoke on cardio-respiratory hospitalizations in San Diego and Tijuana, using the October 2007 wildfires as a case study. The October 2007 wildfires involved about 30 wildfires that began around October 20th in the Southern California coastal region, with the largest area burned concentrated in San Diego County, including areas surrounding the international border [25]. These extreme wildfire events were mainly driven by severe drought and unusually strong Santa Ana winds and, at the time, were considered one the most devastating wildfire events in the history of California. To this day, the largest wildfire during the October 2007 events (Witch Fire) remains the second most destructive fire in Southern California [26]. The San Diego-Tijuana region was completely covered by smoke starting October 20th, 2007. As smoke produced from wildfires in the Southern California-Baja California area can affect populations in the entire region, understanding the burden of these events is critical in addressing this global health concern.

## Methods

### Study context

The San Diego-Tijuana border region is between a Mediterranean and semi-arid climate [27]. Most of the annual precipitation occurs in the winter, from November to March, and is subject to high variability, driving increasingly frequent droughts and floods [28]. The region is affected by Santa Ana winds (SAWs), dry down-sloping winds rooted in cold air masses over the elevated Great Basin, which affect the region primarily between September and May [29]. SAWs are linked to the ignition and spread of wildfires, and spreading the smoke burden during these events [30]. The October 2007 wildfires were a large record-breaking event, which ignited thirteen different wildfires from Los Angeles to Tijuana; this was exacerbated by SAWs [31]. This event as well as other more recent wildfires in the region highlight the vulnerability of the region to this climatic threat. The area has experienced many major wildfire events in recent years, and this is projected to continue increasing in the context of climate change.

Numerous studies have been conducted to understand the epidemiological effects of extreme heat and wildfire smoke on California as a whole [15, 32–34] and in San Diego specifically [16, 17, 35]. Contrastingly, little to no research has been conducted in Tijuana or Baja California to understand these impacts. The proximity of these cities and the discrepancy in available evidence on this topic highlights the need to find strategies to adapt research tools and methods to each context; understanding the impacts of extreme weather events in regions with limited epidemiological data will become increasingly important.

In San Diego and California, it has been well established that wildfire smoke drives adverse health outcomes [15, 17, 36–40]. An evaluation of the health effects of a major wildfire event that occurred in Southern California in 2003 showed increased eye and respiratory symptoms,

medication use, and physician use for children in communities in Southern California [38]. Recent work in Southern California has shown that $PM_{2.5}$ from wildfire smoke can drive up to 10 times more respiratory hospitalizations than non-wildfire $PM_{2.5}$ [15]. A study specific to San Diego County found a 30% (95% CI: 26.6% to 33.4%) increase in emergency and urgent care visits at Rady's Children's Hospital network for each 10-unit increase in $PM_{2.5}$ from wildfire smoke [41]. A study evaluating the economic impact of the 2007 wildfires showed that it drove medical costs by over $3.4 million [36]. These are a few of the numerous studies in San Diego and California that demonstrate the health impacts of wildfire smoke and highlight the need to inform policies to limit the harmful effects of this exposure.

The strong health effects of wildfire smoke observed in San Diego County indicate the need to understand and reveal and compare these impacts in Tijuana. Studying these effects in Tijuana is critical to informing early warning systems and addressing this research gap. Ideally, forecasting systems for wildfire smoke will consider environmental risk factors on both sides of the border and inform warning systems and action plans to protect populations in the entire region.

## Overview of analytical strategy

To consider the effect of wildfire smoke on cardio-respiratory hospitalizations, synthetic control methods (SCM) [42, 43] were applied, capitalizing on the timing of the wildfire as a natural experiment. Although these methods were first developed for econometrics, they have recently been applied to study the effects of acute environmental stressors such as wildfire smoke [43, 44]. Synthetic control methods use the trend in the outcome in the "treated" unit or in this case the geographic region exposed to wildfire smoke, to identify and weight control units that can represent a counterfactual trend of what would have happened if the wildfire smoke had not occurred in the region. The benefit of this approach is that it capitalizes on the temporality of the treatment or wildfire event to compare the pre-treatment and post-treatment hospitalization count by estimating what would have occurred if the wildfire smoke had not hit the region through the identification of synthetic controls. The trend in hospitalizations is then followed after the treatment and any difference between the treated unit and its synthetic control can be attributed to the wildfire smoke. The Municipality of Tijuana and the County of San Diego were the two treated units of interest, while all municipalities in Mexico without wildfire smoke during this period were considered as potential controls for Tijuana, and other counties in California without wildfire smoke during this period were considered as potential controls for San Diego. Two analyses were conducted considering San Diego County and the Municipality of Tijuana separately. The pre-treatment period used to identify the control units was 10 days before the wildfire smoke began to give sufficient days to identify a trend in hospitalization counts. The outcome was followed for 6 days from the first day of the wildfire smoke exposure (October 20th, 2007) to consider the full duration of smoke days above the 30th percentile on both sides of the border.

## Data sources

The Hazard Mapping Smoke product of the National Oceanic and Atmospheric Administration (NOAA) was used to identify smoke plumes for all California counties and Mexican municipalities during the study period (October 11th- October 26th, 2007). This product applied algorithms from visible imagery from various satellites to identify smoke plumes and was revised and modified by trained analysts [45]. In our main analysis, any day for which 30% or more of a county or municipality was covered in smoke was considered to be exposed and any potential control with an exposed day during the study period was not eligible;

**Table 1. Descriptive statistics of study population and hospital admission characteristics for the Municipality of Tijuana and County of San Diego before and after the wildfire smoke event.**

| | San Diego (SD) County | | Municipality of Tijuana (TJ) | |
|---|---|---|---|---|
| Total population (2010 census) | 3,095,313 | | 1,603,955 | |
| Poverty rate (2010 census) | 9.5% | | 32.8% | |
| Education high school graduate or higher[a] | 88% | | 16.7% | |
| Unemployment rate (2010 census) | 10.8% | | 2.4% | |
| Gross Domestic Product (in million dollars) | 171,568 | | 26,721[b] | |
| **Hospital admissions** | **Before smoke (Oct 11th-20th)** | **After smoke (Oct 21st-26th)** | **Before smoke (Oct 11th-20th)** | **After smoke (Oct 21st-26th)** |
| Daily average hospitalization count | 117.5 | 143.4 | 1.5 | 4.2 |
| Hospitalization rate (per million) | 37.9 | 46.3 | 0.9 | 2.6 |

[a]SD for age 25+, TJ age 18+

[b]for the entire state of Baja California

sensitivity analyses were also run considering 20%, 50%, and 70% smoke plume coverage. Visualization of satellite imagery for the border region was obtained from the National Aeronautics and Space Administration Worldview visualization tool [46].

Data on cardio-respiratory hospital admissions for California were obtained from the Office of Statewide Health Planning and Development Patient Discharge Data now renamed to the California Department of Health Care Access and Information Patient Discharge Data [47]. For Mexico, data on cardio-respiratory hospital admissions was obtained from the Mexico Secretary of Health [48]. Any hospital admission with a primary diagnosis code for diseases of the circulatory system (ICD-9: 390–459, ICD-10: I00-I99) and diseases of the respiratory system (ICD-9: 460–519; ICD-10: J00-J99) based on the International Classification of Diseases, Ninth, and Tenth Revision were considered [49, 50]. Although the population of the Municipality of Tijuana is much smaller than San Diego County (1.6 million vs. 3 million), the hospitalization rate remains much higher in San Diego with an average daily case of 40.6 per million during the study period and 1.4 cases per million in Tijuana (Table 1). The hospitalization rate almost 30 times higher in San Diego is driven by differences in the quality and generalizability of the data sources. While the data from San Diego is comprehensive of all patients admitted to hospitals in the County, the data from Mexico comprises only patients admitted at hospitals administered by the public sector through the Secretary of Health [51]. This limits the generalizability of our findings in Tijuana, and we are limited by the availability of health data- to our knowledge, this is the only available dataset that included the date of admission which is required for this analysis. A daily hospital admission count was estimated for each county and municipality and a two-day rolling average was estimated to increase smoothness in the trend when identifying suitable controls.

## Analysis

Hospitalization counts using 2-day rolling averages were used to identify and estimate the synthetic controls using the pre-treatment period (October 11th-20th); SCM accounts for population size by identifying municipalities/counties with similar baseline hospitalization counts. A parametric generalized synthetic control approach was applied which imputes counterfactuals for each San Diego and Tijuana with eligible control groups using a linear interactive fixed effects model with unit-specific intercepts [52]. SCM then follows the trend and computes an average treatment effect in the treated (ATT) by estimating a difference between the treated

units and its synthetic controls for 5 days following smoke exposure (October 20th-26th, 2007). Confidence intervals were estimated using bootstrapping with 500 runs and visualized at an alpha = 0.05. A percentage increase in hospitalizations attributable to wildfire smoke was estimated by taking the ATT and dividing it by the average hospitalization count for the treated unit in the pre-treatment period; due to differences in the data being used in Tijuana and San Diego, this relative change was thought to be a more suitable comparison. A sensitivity analysis was conducted including mean temperature as a covariate estimated from the Oregon State University PRISM Climate Group for California counties using population-weighted centroids and an average of population-weighted daily minimum and maximum temperature estimated using Daymet V4 product and WorldPop for Mexican municipalities [53–55]. All analyses were conducted using R 4.1.0 analytical software [56]. The study protocol was certified as exempt from IRB review from the University of California San Diego Human Research Protections Program (Project #201534XX) and approved by the Bioethics committee of the Faculty of Health Science of the Autonomous University of Baja California.

## Results

During the study period (October 11th-October 26th, 2007), there were 2009 hospital admissions in San Diego County, 1215 for circulatory diagnoses and 794 for respiratory diagnoses. In contrast, the Municipality of Tijuana had only 37 hospital admissions, 21 of which were for cardiovascular diagnoses and 16 for respiratory diseases. The hospitalization rate is higher in the after-smoke period than the before-smoke period in both San Diego and Tijuana (Table 1). Age and gender distribution of hospitalizations for San Diego and Tijuana are shown in S1 Table.

Wildfire smoke from the various 2007 Southern California wildfires (including the Witch, Harris, Poomacha, and Rice wildfires in San Diego County) started impacting the San Diego-Tijuana region on October 21st, 2007. Perimeters of the wildfires are shown in S1 Fig. On October 25th, the entire County of San Diego and Municipality of Tijuana were 100% covered by smoke exposure according to the HMS smoke plumes (Fig 1). In California, 17 counties had no days with 30% smoke coverage during the study period and were eligible as potential controls. In Mexico, 76 municipalities were eligible as potential controls for Tijuana after excluding any municipality with zero hospitalizations for any day during the study period.

The generalized synthetic control estimated a suitable counterfactual trend for both San Diego and Tijuana with an average of less than 9 hospital admissions difference for San Diego and less than 1 case difference for Tijuana during the pre-treatment period (S2 Table). Control units and their weights from the generalized synthetic control approach are provided in the appendix (S3 Table). An increase in hospitalizations was observed following wildfire smoke exposure in both San Diego and Tijuana, although a precise effect was only observed in San Diego and no effect was observed in Tijuana (Fig 2, S2 Fig). Results from sensitivity analyses considering varying percentages of smoke coverage and exposure levels showed similar results (S3 Fig). The ATT on the absolute scale was 0.63 daily cases per million in Tijuana while 6.38 daily cases per million in San Diego for six days following the beginning of the wildfire smoke exposure in the region (Table 2). When including temperature as a covariate, similar results were observed with a precise effect of smoke in San Diego but no effect was observed in Tijuana (S4 Fig). For the sensitivity analysis in Tijuana including municipalities with zero cases on any day during the study period, similar results were observed (S5 Fig).

## Discussion

While the San Diego-Tijuana border is a delineator for differing economic and social conditions, the October 2007 wildfires that started in California brought damaging smoke exposure

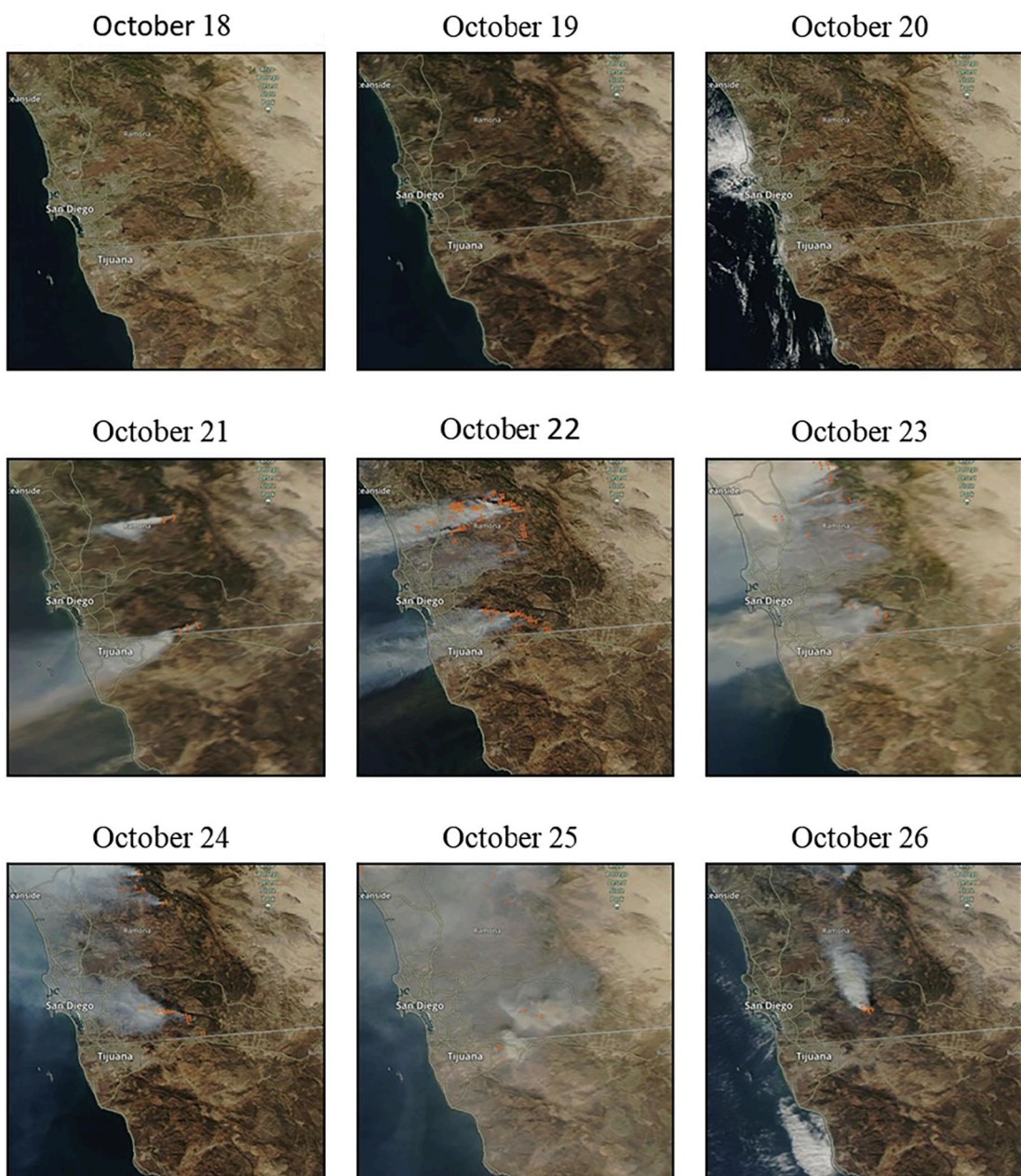

**Fig 1. Map of study region.** San Diego Tijuana border region showing smoke plumes when smoke started covering region during study period (October 18th-26th, 2007). Map data from OpenStreetMap and visualized by NASA worldview [46].

to populations residing on both sides. Capitalizing on the random timing of the start of wild-fire smoke and its spread to the entire border region, we applied synthetic control methods to explore the role of wildfire smoke in driving adverse health outcomes. We observed a positive ATT for the effect of wildfire smoke on cardio-respiratory hospitalizations in both San Diego and Tijuana (Fig 2). The effect, however, differed on each side of the border; while San Diego showed a precise effect of wildfire smoke, we could not confirm an effect of wildfire smoke in Tijuana at the 95% confidence level although results suggest an increase in hospitalizations following smoke exposure (Fig 2).

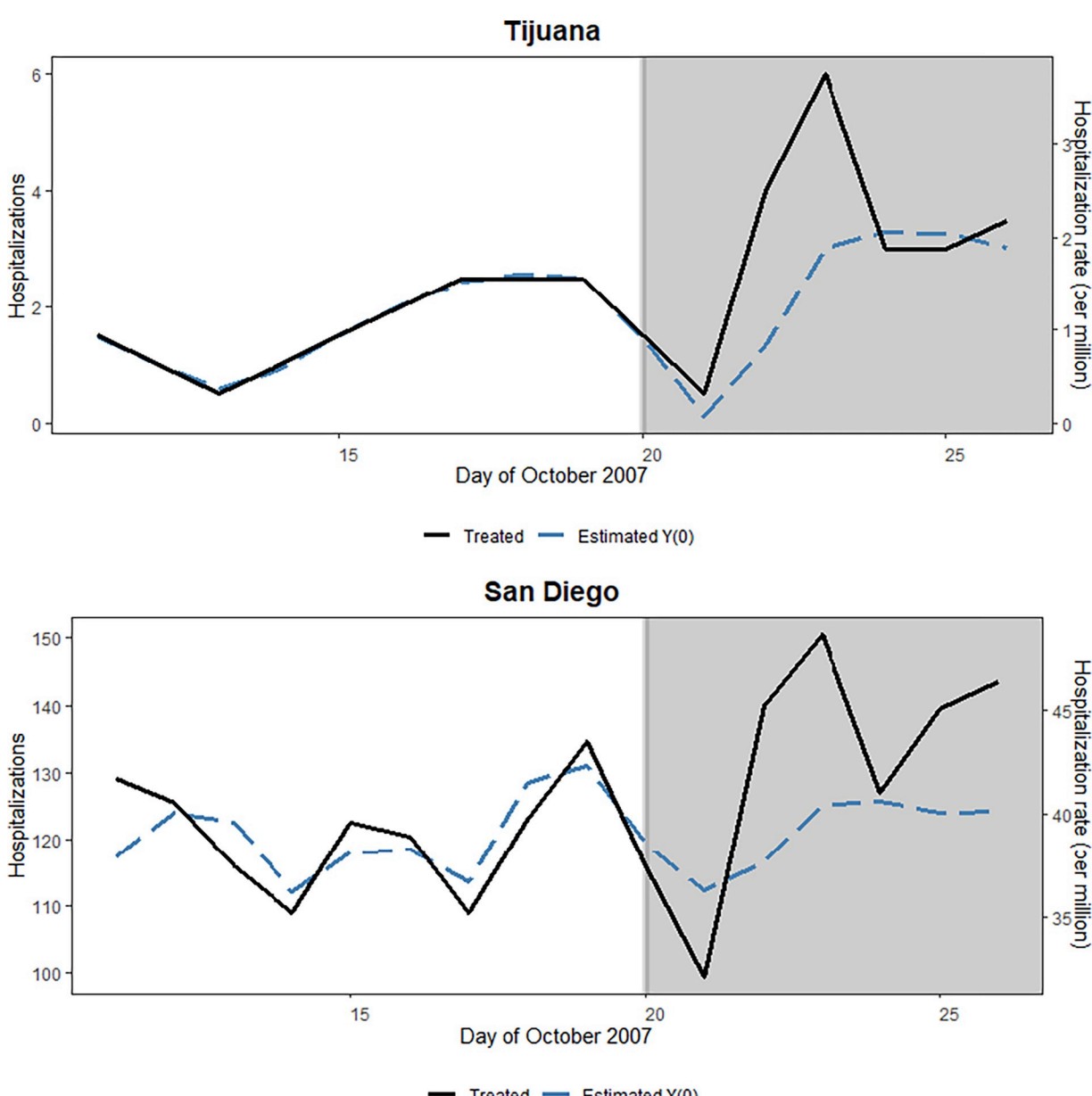

**Fig 2. Results of synthetic control method.** The effect of wildfire smoke on cardio-respiratory hospitalizations in San Diego and Tijuana using synthetic control methods.

**Table 2. Results of synthetic control method.**

|  | San Diego (SD) County | Municipality of Tijuana (TJ) |
|---|---|---|
| Relative (%) change in ATT from smoke exposure overall | 10 | 60.1 |
| Absolute change in ATT and 95% confidence interval | 19.75 [17.69, 21.81] | 1.08 [-6.87, 9.02] |
| Absolute change in ATT per million | 6.38 | 0.67 |

Change in the average treatment effect in the treated (ATT) on cardio-respiratory hospitalizations in San Diego and Tijuana from the onset of smoke exposure using synthetic control methods

These results indicated a differential susceptibility to the effects of wildfire smoke. The social context of San Diego and Tijuana differ- in 2010, the poverty rate was 3 times higher in Tijuana than San Diego while the percentage of the population with a high school education was 5 times higher in San Diego according to the 2010 census in both countries (Table 1). Socio-demographics including lower education and employment rates have been shown to increase vulnerability to environmental hazards such as air pollution, as people with higher educational attainment may have higher income, improved access to healthcare and may have improved knowledge to manage health risks [57]. Furthermore, the Gross Domestic Product (GDP) of San Diego County was over $170,000 million while it was just over $26,000 million for the entire state of Baja California Norte in 2010 [6, 58]; this could also play a role in accessibility to economic and healthcare services. Additionally, Tijuana has one of the highest crime rates for cities in Mexico, which may contribute to increased social vulnerability [59]. Other harmful environmental exposures such as higher traffic related pollution can also increase susceptibility to wildfire smoke, which is highly relevant in the this region where traffic related pollution is high due to idling vehicles at the border [60]. Although social conditions are hypotheses for which differing effects could be observed, additional research would have to consider the role of social vulnerability to confirm this as a proposed mechanism in the context of wildfire smoke.

When considering the percentage change in hospitalizations following wildfire smoke exposure, the relative increase in hospitalizations in Tijuana was greater than observed in San Diego, with an increase six times higher although no effect was detected at the 95% confidence level (Table 2). Lower socio-economic status increases the risk of air pollution on various health outcomes [19–21], and there are many socio-economic factors that could be driving an increased susceptibility in Tijuana. The poverty rate of Tijuana is more than three times that of San Diego and less than 25% of the population in Tijuana have completed a basic education while this percentage is more than three times higher in San Diego (88%) (Table 1). The unemployment rate, however, is higher in San Diego but this is probably driven by differences in government programs that provide social services and collect data on this population. Not only do Mexico and the US have differing socio-economic profiles at the national level, but even within each country, inequality intensifies. Out of all cities in Mexico, Tijuana is ranked to have the fourth worst quality of life index based on geographical, environmental, and social factors [61]. In contrast, San Diego is ranked one of the richest cities in the U.S. [62]. The socio-economic differences are likely playing an important role in the potential increased susceptibility to wildfire smoke observed in Tijuana, although the data quality challenged the ability to detect a precise effect of smoke in this analysis.

Interestingly, we observe a drop in the 2-day moving average of cardio-respiratory hospitalizations immediately succeeding the start of the wildfire smoke followed by a steep increase in both San Diego and Tijuana. One possible reason for this could be that there are behavioral changes related to the onset of a high smoke episode that could increase the protection of the population from air pollution and other hazards but which don't last more than one or two days. The implications of this immediate decrease should be further explored to investigate what specific actions and information are the most effective in protecting individuals from wildfire smoke.

There are many actions that can be activated in the context of a wildfire smoke event and may be harmonized in a border region to best protect population health. Warning systems to forecast wildfire smoke exposure have the potential to decrease its burden; smoke-related public health messaging to stay indoors can prioritize at-risk populations [63]. Although there is a need for more research to evaluate the effectiveness of these measures on protecting population health, understanding which characteristics make a neighborhood or area susceptible to

the effects of wildfire smoke is critical to the activation of any intervention to maximize public health benefits. Additionally, this could be a factor in explaining the differential impacts observed as residents of San Diego County were likely notified of the wildfire events from evacuation warnings and advisories while Tijuana may have not received information since fires were burning primarily on the US side of the border. This highlights the need for harmonized warning systems to protect populations from the health effects of this exposure. Although programs for transborder collaboration have been implemented, unfortunately, they usually lack institutionalization and long-term continuity [64]. Mechanisms should be formalized to jointly develop and implement actions between agencies in both San Diego and Tijuana to implement a binational response to environmental health hazards such as wildfire smoke. For example, the California-Baja California 2019–2020 Border 2020 Action Plan includes a Joint Contingency Plan for environmental response to chemical hazardous substances; the mechanisms put in place with this program could be adapted to respond to wildfire smoke events [65].

To our knowledge, neither San Diego nor Tijuana has an established early warning system or action plan to protect populations during these events [66]. The Air Quality Index can be used to evaluate local air and regional air quality conditions during a wildfire event, but no specific communication or intervention is activated based on these exposure levels [66]. Strengthening the network of environmental health research in Baja California and San Diego can help increase the preparedness of the region in the context of climate change. Forecast systems for SAWs have been shown to be accurate for a 6–7 day lead time [67], which can be useful in the prediction of periods of high wildfire risk. Wildfire and heat wave forecasting systems could be used by policymakers to develop and implement actions to limit the morbidity and mortality attributable to extreme weather in this region. Ideally, integrated early warning systems that account for local environmental measures and predict population exposure to wildfire smoke will be implemented to protect public health on both sides of the border by taking into account local epidemiologic information.

There is a major disparity in evidence that we observe on a global scale with the majority of evidence on the epidemiological effects of wildfire smoke coming from the US and Australia [13] and very few studies estimating these effects in low and middle-income countries [18]. Unfortunately, the limited data accessibility and quality is one of the drivers for the little evidence coming from low and middle income countries. Poor data quality and availability are often a barrier to health research in these regions; lack of data standards can further contribute to this challenge [68]. Although data from resource-constrained regions may be more limited, studying the effects of wildfire smoke in these settings is critical to understanding and addressing these impacts.

It is important to interpret the findings of this research in light of certain limitations. First, there could be some exposure misclassification as the hazard mapping tool used to identify smoke plumes does not differentiate between smoke at the ground-level and smoke higher up in the troposphere. Also, smoke coverage percentages are considered a proxy for the population exposure, therefore exposure misclassification may remain. However, previous studies estimating wildfire-specific $PM_{2.5}$ in San Diego County confirmed high levels of particulate matter from wildfires during this period indicating that exposure misclassification may be minimal during this particular event [15]. Also, it is important to consider that Tijuana and San Diego have differing population structures and San Diego County has an older population demographic than the Municipality of Tijuana. It is also important to note that there may be cross-border health care seeking in a border context that could alter the number of patients on either side of the border; it has been shown that many immigrants return to Mexico to seek out healthcare [69], which could affect our datasets and findings. A study evaluating these

health seeking behaviors on both sides of the border would be valuable to contextualize our findings but this is beyond the scope of this work. Lastly, the hospitalization data in Tijuana is not generalizable, limiting the potential conclusions and comparisons that can be drawn from this analysis. As the hospitalization dataset from Mexico is from the public health system, it may be biased as the population served may be more susceptible. However, we felt this analysis was valuable using the available data even if it is not the ideal comparison.

Although the socio-demographic contexts of San Diego and Tijuana are very different, it is important to consider these two cities as one border region that undergoes similar climate challenges that will continue to be exacerbated under climate change. The importance of developing a concerted effort to protect populations on both sides of the border from these adverse health effects will become increasingly critical. Wildfires will continue to ignite and spread to the region and with more frequency and severity [70]. There is a strong need for policy-relevant evidence to inform interventions to protect populations in regions such as the San Diego-Tijuana border that are highly vulnerable to wildfires.

In conclusion, these results can be used to understand how the effects of environmental exposure can differ across a socio-demographically diverse border region. This approach highlights how we can capitalize on border regions to explore the differential effects of environmental health risks that spread beyond borders. We hope that this can be applied to other border regions to continue to explore what drivers increase susceptibility to environmental health hazards.

## Supporting information

**S1 Fig. Map of 2007 wildfires perimeter.** Spatial extent of Witch, Harris, Poomacha, and Rice and Ammo wildfires that burned in San Diego County, October 2007, map data from OpenStreetMap [71].
(PDF)

**S2 Fig. Results of synthetic control method with confidence intervals.** The effect of wildfire smoke on cardio-respiratory hospitalizations in San Diego and Tijuana using synthetic control methods with confidence intervals.
(PDF)

**S3 Fig. Results of sensitivity analyses using varying smoke coverage levels.** Sensitivity analyses considering 20%, 50% and 70% smoke coverage as exposure for San Diego and Tijuana.
(PDF)

**S4 Fig. Results of sensitivity analysis with temperature.** Results of sensitivity analysis including daily mean temperature as a covariate in evaluating the effect of wildfire smoke from October 2007 wildfires on cardio-respiratory hospitalizations in San Diego County and the Municipality of Tijuana.
(PDF)

**S5 Fig. Results of sensitivity analysis including additional controls for Tijuana.** Results of sensitivity analysis including all potential controls (not excluding municipalities that had any day with 0 cases) of the effect of October 2007 wildfire smoke on cardio-respiratory hospitalization in the Municipality of Tijuana.
(PDF)

**S1 Table. Demographics of hospitalizations.** Gender and age distribution of hospitalizations in San Diego County and the Municipality of Tijuana during study period (October 11th-26th,

2007).
(DOCX)

**S2 Table. Difference between treated and counterfactual.** Difference between the daily cases in each treated unit and its counterfactual for the pre-treatment period.
(DOCX)

**S3 Table. Weights for the synthetic control methods.** Controls and their weights estimated using generalized synthetic control methods.
(DOCX)

## Acknowledgments

This project was supported by the Fogarty International Center of the National Institutes of Health (NIH) under Award Number D43TW009343 and the University of California Global Health Institute (UCGHI). The content is solely the responsibility of the authors and does not necessarily represent the official views of the NIH or UCGHI. The authors would also like to thank Pedro Antonio Ramonetti Vega and Gordon McCord for their support in data curation and cleaning.

## Author Contributions

**Conceptualization:** Lara Schwarz, L. C. Aguilar-Dodier, Javier Emmanuel Castillo Quiñones, María Evarista Arellano García, Tarik Benmarhnia.

**Data curation:** Lara Schwarz, Rosana Aguilera.

**Formal analysis:** Lara Schwarz, Rosana Aguilera.

**Funding acquisition:** Lara Schwarz, María Evarista Arellano García, Tarik Benmarhnia.

**Investigation:** Lara Schwarz.

**Methodology:** Tarik Benmarhnia.

**Supervision:** María Evarista Arellano García, Tarik Benmarhnia.

**Writing – original draft:** Lara Schwarz.

**Writing – review & editing:** Lara Schwarz, Rosana Aguilera, L. C. Aguilar-Dodier, Javier Emmanuel Castillo Quiñones, María Evarista Arellano García, Tarik Benmarhnia.

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
