## [Decision Letter · Decision Letter 0]

13 Mar 2023

PGPH-D-23-00161

Wildfire smoke knows no borders: differential vulnerability to smoke effects on cardio-respiratory health in the San Diego-Tijuana region

Dear  Ms Lara Schwarz

Thank you for submitting your manuscript to PLOS Global Public Health. After careful consideration, we feel that it has merit but does not fully meet PLOS Global Public Health’s publication criteria as it currently stands. Therefore, we invite you to submit a revised version of the manuscript that addresses the points raised during the review process.

We look forward to receiving your revised manuscript.

Kind regards,

Reginald Quansah, Ph.D.

Academic Editor

Journal Requirements:

Additional Editor Comments (if provided):

Reviewers' comments:

Reviewer's Responses to Questions

**Comments to the Author**

1. Does this manuscript meet PLOS Global Public Health’s publication criteria? Is the manuscript technically sound, and do the data support the conclusions? The manuscript must describe methodologically and ethically rigorous research with conclusions that are appropriately drawn based on the data presented.

Reviewer #1: Partly

Reviewer #2: Yes

Reviewer #3: Yes

Reviewer #4: Partly

2. Has the statistical analysis been performed appropriately and rigorously?

Reviewer #1: I don't know

Reviewer #2: Yes

Reviewer #3: Yes

Reviewer #4: Yes

3. Have the authors made all data underlying the findings in their manuscript fully available (please refer to the Data Availability Statement at the start of the manuscript PDF file)?

Reviewer #1: No

Reviewer #2: Yes

Reviewer #3: Yes

Reviewer #4: Yes

4. Is the manuscript presented in an intelligible fashion and written in standard English?

Reviewer #1: No

Reviewer #2: Yes

Reviewer #3: Yes

Reviewer #4: Yes

5. Review Comments to the Author

Reviewer #1: The concern of wildfire smoke and its effects on cardio-respiratory health is of significance. However, I am not the expert to comment on the captioned article. And therefore, I would leave with no comment nor recommendation for this article.

Reviewer #2: This study aimed to examine the differential effect of wildfire smoke on cardio-respiratory hospitalizations in San Diego and Tijuana using data from 2007 wildfires hospitalization. The novelty of the study lies in the use of spatial data with health data to examine this differential effect and the rigor of statistical modeling. However, I have some concerns that need clarification from the authors.

1. The introduction is well written, and the objectives of the paper clearly stated.

2. The current description of the October 2007 wildfires provides a good context and motivation to the study; however, I think it can be expanded. You could create a sub-section for the study context. Highlight the prevalence of wildfires in the area, their effect, studies related to the health effects of wildfires, and why the focus on the 2007 wildfires. You could also talk about the duration and the impact it had. Then narrow down to focus on the cardio-respiratory infections that may have been overlooked.

3. The authors adequately describe the two forms of data used in the analysis but there is little information on how the data were merged and the level at which these data were examined. How did the authors adjust for spatial autoregression?

4. While spatial data provide a good proxy for measuring particulate matter concentration, the authors may want to consider objectives measures of particulate matter concentrations in the area at the time. Where inaccessible, the authors may want to highlight the limitation of using only spatial data in assessing exposure to PM2.5

5. It is less evident in the paper that the socio-demography of San Diego and Tijuana were considered in the computation of the models although some description of it is given in the results and discussion. From the current description, I do not see an adjusted spatial model. Kindly consider providing some information of this in the paragraph on data analysis.

6. Please give some time to proofread the manuscript and correct all typos (there are few that need editing).

Reviewer #3: This is a very interesting article on the nexus between wildfire smoke and cardio-respiratory diseases. Authors have revealed important role of socio-demographics in modulating the effects of wildfire smoke and can be potentially useful in developing a concerted regional effort to protect populations on both sides of the border from the adverse

health effects of wildfire smoke. Findings are interesting. However, I suggest to revisit conclusion of the study and make it specific with concreate recommendations. I also suggest researchers to state limitations of the study in discussion section.

Reviewer #4: This is an interesting and important study looking at cross-border health effects of wildfire smoke exposure using the indicator of hospitalizations. While the data is more limited from Tijuana and this may affect the conclusions, it is still very important to conduct these types of studies and not leave out LMIC contexts because of more limited data.

Page 7: The last line is missing a word that affects clarity (is it meant to say “is driven by” or “drives”)? “The hospitalization rate almost 30 times higher in San Diego is driven by differences the quality and generalizability of the data sources.”

Table 1: GDP is reported in millions of dollars (so would be in $1+ trillion for San Diego and $26+ million in Tijuana) but on page 12, it states “Furthermore, the Gross Domestic Product (GDP) of San Diego County was over $1.2 million while it was just over $26,000 for the entire state of Baja California in 2010 (7).” These are very different and should be clarified which is correct (or if either are correct).

Page 13: This is a strong statement that is not supported by the methods or statistical analysis: “The socio-economic differences are likely playing an important role in increased susceptibility to wildfire smoke observed in Tijuana.” The increased hospitalizations in Tijuana, while the relative change is higher, is not statistically significant (possibly due to small sample size of hospitalizations there). In addition, the methods did not assess for a relationship between individual and neighborhood-level SES and hospitalizations or air pollution exposure, so this is really conjecture. While it is an important hypothesis and structural factors are likely influencing exposure to air pollution and health effects from this exposure, the data does not support this. In reality, the data shows that there is a statistically significant change in hospitalizations in San Diego, but not in Tijuana – there are other potential explanations for which that could be the case, including higher baseline co-morbidities in San Diego population (this is a hypothesis/example).

Page 13: “Warning systems to forecast wildfire smoke exposure have potential to decreasing the burden; smoke-related public health messaging to stay indoors can prioritize at-risk populations (45).” I think this is supposed to say, “have potential to decrease”

Pg 14-15: Would add to limitations the much smaller number of hospitalizations in Tijuana and how this many have impacted the data. In particular, there may be skewed data there given that hospitalizations were only noted in public hospitals, which presumably may serve lower-income and more vulnerable patients. This could exaggerate the finding of relative increased hospitalizations during the study period in this population.

6. PLOS authors have the option to publish the peer review history of their article (what does this mean?). If published, this will include your full peer review and any attached files.

**Do you want your identity to be public for this peer review?** For information about this choice, including consent withdrawal, please see our Privacy Policy.

Reviewer #1: No

Reviewer #2: No

Reviewer #3: **Yes: **Meghnath Dhimal

Reviewer #4: No

---

## [Editor Report · Decision Letter 1]

2 Jun 2023

Wildfire smoke knows no borders: differential vulnerability to smoke effects on cardio-respiratory health in the San Diego-Tijuana region

PGPH-D-23-00161R1

Dear Ms Schwarz,

We are pleased to inform you that your manuscript 'Wildfire smoke knows no borders: differential vulnerability to smoke effects on cardio-respiratory health in the San Diego-Tijuana region' has been provisionally accepted for publication in PLOS Global Public Health.

Best regards,

Reginald Quansah, Ph.D.

Academic Editor